# Low Sleep Satisfaction Is Related to High Disease Burden in Tinnitus

**DOI:** 10.3390/ijerph191711005

**Published:** 2022-09-02

**Authors:** Franziska C. Weber, Winfried Schlee, Berthold Langguth, Martin Schecklmann, Stefan Schoisswohl, Thomas C. Wetter, Jorge Simões

**Affiliations:** 1Department of Psychiatry and Psychotherapy, University of Regensburg, 93053 Regensburg, Germany; 2Interdisciplinary Tinnitus Centre, University of Regensburg, 93053 Regensburg, Germany; 3Department of Psychology, Bundeswehr University Munich, 85577 Neubiberg, Germany; 4Center for Sleep Medicine, University of Regensburg, 93053 Regensburg, Germany

**Keywords:** tinnitus, tinnitus disorder, tinnitus distress, depression, sleep, sleep disturbances, sleep satisfaction, mental health, quality of life, multinominal regression model

## Abstract

Previous studies have shown a high prevalence of sleep disturbances in tinnitus patients. However, no study has yet evaluated subjective sleep satisfaction. The present study aimed to investigate associations of self-reported sleep satisfaction with sociodemographic factors, tinnitus-related distress, depression, and self-reported quality of life. This is a retrospective analysis of 2344 outpatients with tinnitus presenting at a tertiary German tinnitus clinic from 2010 to 2020. Patients who filled in five questionnaires (Tinnitus Handicap Inventory (THI), Tinnitus Questionnaire (TQ), Major Depression Inventory (MDI), Tinnitus Sample Case History Questionnaire (TSCHQ), and the World Health Organization Quality of Life Brief Version (WHOQOL-Bref)) were included. Based on the question about sleep satisfaction in the WHOQOL-Bref, group classification into (I) sleep-satisfied, (II) neither satisfied or dissatisfied, and (III) sleep-dissatisfied patients was performed. Associations between sleep satisfaction and quality of life, depression, tinnitus distress, and tinnitus characteristics were analyzed by group differences and a multinomial regression model with elastic net penalization. A total of 42.38% of patients were satisfied or very satisfied with sleep, whereas 40.91% of patients were dissatisfied or very dissatisfied with sleep. The remaining patients reported being neither satisfied nor dissatisfied with sleep. Sleep-dissatisfied patients were significantly more burdened in questionnaires on depressive symptoms (MDI), tinnitus distress (TQ, THI), and quality of life (WHOQOL-Bref). In addition, they suffered significantly more often from comorbidities such as headache, neck pain, or temporomandibular joint disorder (TMJ). The elastic net regression based on sum scores of THI, TQ, MDI, the four domains of WHOQOL-Bref, as well as all individual questions from the TSCHQ was able to classify patients satisfied with their sleep with an accuracy of 79%, 87.8% sensitivity, and 70.4% specificity. The model could not identify patients indifferent with the quality of their sleep (neither satisfied nor dissatisfied) (sensitivity: 0%; specificity: 100%). The accuracy of the model to predict patients dissatisfied with their sleep was 80.7%, with 83% sensitivity and 78.4% specificity. Poor physical and mental health (Domain I/II WHOQOL-Bref) as well as tinnitus distress were the strongest predictors of sleep dissatisfaction. Conversely, for sleep satisfaction, good physical and mental health as well as low tinnitus distress were the strongest predictors. The division into sleep-satisfied and sleep-dissatisfied tinnitus patients allows a very good discrimination regarding disease burden as indicated by depression, tinnitus distress, quality of life, and pain-related comorbidities. Physical and mental health as well as tinnitus distress seem to be strongly related to sleep satisfaction underscoring the concept of “tinnitus” versus “tinnitus disorder”, but also the importance of sleep satisfaction as a global health indicator. Moreover, these data indicate the relevance of addressing sleep disorders in the therapeutic management of chronic tinnitus patients.

## 1. Introduction

Tinnitus is the perception of a sound in the absence of an external acoustic sound source [1,2,3]. With a prevalence of 10–15%, it is a common disorder, with a total of about 1–2% of all people experiencing relevant suffering with significant impairment of quality of life [4,5,6,7]. The incidence of tinnitus is increasing according to a recent large epidemiological study, with exposure to noise being identified as a main cause [8]. Overall, there is a wide range of clinical manifestations, e.g., regarding loudness, pitch, sound quality, laterality, maskability, or comorbidities [2]. Comorbidities such as hyperacusis [9], hearing loss [10,11], sleep problems [12,13,14,15,16], depression [17,18], and pain disorders [19] can have a major impact on the level of suffering and thus also on the quality of life [20].

To date, no causal therapeutic options are available for patients with tinnitus to reliably suppress the phantom perception [2,21]. Consequently, adequate treatment of relevant comorbidities is of great importance for the reduction of disease burden in tinnitus patients [22]. For this purpose, it is essential to identify the comorbidities that are associated with distress. Sleep disturbances are very frequently reported by patients with tinnitus and are a major factor reducing patients’ quality of life [20,23].

Many studies of people with tinnitus demonstrate the frequent occurrence of sleep problems in general [24]. Moreover, associations between tinnitus and specific sleep disorders such as insomnia [12,13] obstructive sleep apnea syndrome (OSAS) [25,26,27], nightmares [28,29], or periodic limb movement disorder (PLMD) [30] have been described. The prevalence of sleep disturbance and/or concomitant insomnia ranged from 10.1% to 79.5%, with methodological reasons being blamed for the wide range, such as group selection or assessment of sleep problems [24]. Sleep disturbances have been repeatedly shown to be associated with higher levels of tinnitus distress [13,16,31,32,33,34]. However, the directionality of the relationship is not entirely clear. Is tinnitus causing sleep disturbances, is insomnia a risk factor for tinnitus, or are tinnitus and sleep disorders linked by a third factor [35]? Insomnia and sleep disturbances are also considered a transdiagnostic process in the context of other mental conditions such as depression [36,37]. In this context, a bidirectional relationship between sleep and the original disorder has been shown for disorders such as depression or posttraumatic stress disorder (PTSD) [38,39]. Presumably, the relationship between tinnitus and sleep is also bidirectional. This is supported by the fact that several studies have found that reduction in tinnitus severity was also accompanied by reduction in sleep problems [40,41,42], with cognitive behavioral therapy in particular also proving successful [43]. Disorder-specific cognitive behavioral therapy for insomnia can have a positive effect on tinnitus complaints [44,45]. Furthermore, for the first time, evidence emerged that sleep disturbances prior to the onset of tinnitus can be predictive of the severity of tinnitus [46]. A recent longitudinal study also showed that severe tinnitus was associated with a poorer sleep quality [47]. Overall, the results suggest that sleep problems in tinnitus should not only be considered as an epiphenomenon, but rather as an important factor for the perceived disease burden. In this context, the assessment of sleep satisfaction might be relevant both for clinical management and clinical research.

To our knowledge, no systematic evaluation has been conducted on subjective sleep satisfaction of patients with tinnitus so far. To fill this gap, we retrospectively evaluated questionnaire data from patients who presented at the interdisciplinary tinnitus center at the University of Regensburg. Based on the question about sleep satisfaction of the World Health Organization Quality of Life Bref (WHOQOL-Bref) [48], a group classification into (I) sleep-satisfied, (II) neither satisfied or dissatisfied, and (III) sleep-dissatisfied patients was performed in order to investigate group differences regarding disease burden, comorbidities, and tinnitus characteristics on the one hand and to develop a model predictive for sleep satisfaction in tinnitus patients on the other hand.

## 2. Materials and Methods

### 2.1. Data Collection

Data were used from 2344 patients who visited the tertiary tinnitus clinic between 2010 and 2020 and contributed data to the TRI database [49]. Only assessments from the first visit in the clinic were used, before any therapeutic intervention was performed. This decision was made to mitigate the potential biases of clinical interventions and/or multiple visits to the clinic on the target variable, namely sleep satisfaction. Ethical approval was obtained from the Ethics Committee of the University of Regensburg, protocol number 08/046, and all patients provided their written consent after oral information to have their data anonymously stored, analyzed, and published for scientific purposes.

### 2.2. Variables

Five questionnaires were included in this analysis: tinnitus handicap inventory (THI), tinnitus questionnaire (TQ), major depression inventory (MDI), tinnitus sample case history questionnaire (TSCHQ), and the World Health Organization quality of life brief version (WHOQOL-Bref). These questionnaires are part of the standard clinical evaluation protocol and, therefore, having them all completed and the provision of consent were the sole inclusion criteria adopted in this study.

The THI contains 25 questions, which can be rated on a 3-point scale. The questionnaire was developed as a diagnostic tool to measure tinnitus distress. Its scores range from 0–100, with higher scores representing greater distress, and focus mainly on the impact of tinnitus on daily life [50]. The questionnaire has shown high internal consistency with a Cronbach alpha of 0.94 [51].

Similar to the THI, the TQ is also a psychometrically validated instrument for measuring tinnitus-related distress. Although most of the THI questions are related to psychological aspects of distress, the TQ also encompasses other facets of distress, such as the impact of tinnitus on lifestyle, sleep, communication, and overall health. Each of the 52 questions can be answered with a 3-point scale, yielding total scores ranging from 0–84, with higher scores indicating higher distress. The TQ also has high internal consistency with a Cronbach alpha of 0.95 [52].

The MDI has been developed to quantify major depression, in its mild, moderate, and severe manifestation, according to the DSM-4 and ICD-10 [53,54]. It contains 10 questions, and each question can be answered with a 5-point scale. Its final score is calculated by summing each item, yielding results ranging from 0 to 50 and a Cronbach alpha of 0.94.

The TSCHQ is a tool designed to profile patients with tinnitus in a standardized manner [49,55]. The questionnaire contains 35 questions and focuses on demographics (e.g., age, sex, handedness), tinnitus history and characteristics (e.g., whether other members of the family also suffer from tinnitus, time since tinnitus onset, laterality of the perceiving sounds, type of sound perceived), and comorbidities (e.g., hearing loss, vertigo/dizziness, headaches, temporomandibular joint disorders (TMJ)).

The short version of WHOQOL [48] has 26 questions and comprises four different domains that have an impact on life quality: physical and psychological health, social relationships, and environment. Patients can answer questions with a Likert scale ranging from 1–5 on how they (dis)agree with each statement. Likert scale ranging from 1–5 on how they (dis)agree with each statement, and scores per domain range from 4 to 20. The Cronbach alpha of the four domains of the WHOQOL-BREF are 0.82 (physical health), 0.81 (psychological health), 0.68 (social relationships), and 0.80 (environment) [56].

### 2.3. Data Preparation

Question 16 of WHOQOL-Bref asks ‘How satisfied are you with your sleep?’, which was used as a response variable in the regression analysis. To this end, the options ‘very dissatisfied’ and ‘satisfied’, as well as the options ‘satisfied’ and ‘very satisfied’, were combined, producing three groups: (very) dissatisfied with sleep, neither dissatisfied nor satisfied with sleep, and (very) satisfied with sleep. The sum scores of the THI, TQ, MDI, as well as the four domains of WHOQOL-Bref (i.e., physical health (minus Question 16), psychological health, social factors, and environmental factors), and all individual questions from the TSCHQ, were used as predictors in the regression model (see below).

Predictors were preprocessed prior to fitting the models with the following steps: (1) centering numeric items and scaling numeric items so their means and standard deviation equal 0 and 1, respectively; (2) transformation of categorical variables into dummy variables using one-hot encoding to transform categorical variables into their binary vector representation, and (3) imputing for potential missing values with k-Nearest Neighbor [57] (4) removal of variables with near-zero variance, and (5) merging infrequent categorical variables.

### 2.4. Statistical Analysis and Model Fitting

For group differences with respect to sleep satisfaction, continuous variables are reported as mean (and standard deviation), and median [minimum and maximum values], and categorical variables are reported as count with percentages in parenthesis (%). We used the Kruskal–Wallis test to assess differences between groups for continuous variables, and the χ² chi-squared test for categorical variables. Two-sided *p*-values less than 0.05 were considered statistically significant. All reported *p*-values were corrected for multiple comparisons using the Holm method [58].

For the prediction of sleep satisfaction, we used a multinomial regression with lasso penalization strategy to predict self-reported sleep satisfaction. This model was selected because (1) it minimizes the risk of overfitting by shrinking coefficients to zero to enhance prediction accuracy, a feature especially useful when predictors are correlated; and (2) it provides coefficients that allow inferring about the relation between predictors and the response variables [59].

Data were split into ‘training’ and ‘test’ samples with an 80/20 ratio, that is, the model was trained with 80% of the data, and the final performance was assessed with the remaining 20% of the data [60]. Stratification was used to ensure the same proportion of patients (dis)satisfied with sleep or neither satisfied nor dissatisfied with sleep. The model was trained using a 10-fold cross-validation scheme to identify the optimal tuning parameter λ, which controls the overall strength of the shrinking penalty.

The model with the lowest mean squared error was selected and once the best performing model in the training sample was identified (*n* = 1889), it was applied in the independent test sample (*n* = 455). The reported accuracy, sensitivity, and specificity were obtained by applying the model in the test sample.

### 2.5. Statistical Software

All analysis, figures, and tables were produced in R (version 4.0.3, R Core Team, Vienna, Austria, 2020). Data preparation was performed with the support of the packages tidyverse [61] and with the support of recipes and R-samples packages from tidymodels with its default settings [62]. The elastic net regression was fitted analysis with the tidymodels package [62].

## 3. Results

Table 1 shows the demographics of the 2344 patients included in our analysis, stratified according to self-reported sleep satisfaction. Regarding the distribution of sleep satisfaction, a multinomial distribution of (very) sleep-dissatisfied (40.91%) and (very) sleep-satisfied (41.38%) patients was observed (see Figure 1). The results shown in the table were obtained before data preprocessing (e.g., imputation, center, and scaling). This univariable comparison showed that sex and age did not have a significant effect on sleep satisfaction. Higher depression scores and tinnitus distress, as well as a lower quality of life in the four WHOQOL domains, were associated with self-reported sleep dissatisfaction. Likewise, statistically significant differences were observed between comorbidities and sleep satisfaction, where the presence of comorbidities (e.g., headaches, vertigo/dizziness, TMJ, and neck pain) was associated with poorer sleep satisfaction. Interestingly, variables related to tinnitus characteristics, such as the type of onset, its laterality, or the type of perceived sound type, were not statistically significant different between the three groups. However, pulsatile tinnitus, regardless of whether it was synchronous or not with heartbeat, was associated with sleep dissatisfaction. Constant tinnitus was also significantly associated with sleep dissatisfaction compared to intermittent tinnitus.

Next, we used a 10-fold cross-validated Lasso regression to predict self-reported sleep satisfaction. This way, multivariate analysis of the effects was possible, as well as the classification of sleep satisfaction. The coefficients of the model are presented in Table 2. Values represent the log-odd of being classified into one of the three groups after accounting for the effect of covariates. Dashed lines represent variables that were set to 0, as they did not increase the prediction accuracy of the model. Positive/negative values indicate higher/lower log-odds of being classified into one of the three groups, after accounting for all the other variables. The strongest effects were observed among the physical and psychological health variables of WHOQOL, followed by THI and TQ. From this multivariable comparison, it was observed that poor physical and mental health as well as great tinnitus distress predicted sleep dissatisfaction. Males were less likely to report being (very) dissatisfied with their sleep compared to women. Regarding the characteristics of tinnitus, patients who reported non-pulsatile tinnitus had greater log-odds of being classified in the (very) satisfied group compared to patients whose pulsatile tinnitus are synchronous with the heartbeat. Complex tinnitus sounds (e.g., white noise or ‘cricket-like’ sounds) were also associated with greater sleep satisfaction compared to patients with tonal tinnitus.

The confusion matrix comparing the predicted groups with the real values is presented in Table 3. The model obtained a modest accuracy of 69.5% (95% confidence interval of 65–73.6%), and an R2 of 26%. However, these results were skewed by the model’s inability to correctly predict patients who were neither satisfied nor dissatisfied with sleep. The model could not identify patients neither satisfied or dissatisfied with sleep (0% sensitivity, and 100% specificity). The accuracy of the model to predict patients (very) dissatisfied with their sleep was 80.7%, with 83% sensitivity and 78.4% specificity. For patients (very) satisfied with sleep, the model reached an accuracy of 79%, with 87.8% sensitivity and 70.4% specificity.

## 4. Discussion

### 4.1. Concept of Sleep Satisfaction in Tinnitus

Whereas several previous studies had already investigated the relationship between tinnitus and sleep, here we investigated for the first time the concept of sleep satisfaction in patients with tinnitus. Sleep satisfaction was measured with one question from WHOQOL-BREF. A comprehensive sleep satisfaction survey tool has only been developed recently [63,64] and was not yet available at the time of data assessment for this study. Therefore, no direct conclusions can be drawn in the present study on which individual, social, and environmental factors are relevant for sleep satisfaction or dissatisfaction. It should also be emphasized that the response variable used in this study only reflects the subjective estimation of affected persons and no objective evaluation was performed by instrumental diagnostics such as actigraphy or polysomnography. For future studies, further characterization of sleep satisfaction by validated self-report questionnaires such as the Pittsburgh Sleep Quality Index (PSQI) for subjective sleep quality [65] or the Insomnia Severity Index (ISI) [66,67] for insomnia severity would be beneficial. The strength of the model used here is its simplicity and thus its clinical practicability, in that an orientation classification into satisfied and dissatisfied sleepers is made based on one question. While previous studies of tinnitus patients with screening questions regarding sleep related these to tinnitus (e.g., “Does tinnitus interfere with sleep?”) [68,69], the screening question here was asked independently of the tinnitus. As already reviewed in the introduction, there is increasing evidence that sleep disturbances in patients with tinnitus should not only be considered as an epiphenomenon, but also as an important influencing factor. Should the postulated bidirectional relationship of tinnitus and sleep be corroborated, sleep difficulties independent of tinnitus should be focused on in addition to the sleep items in validated tinnitus questionnaires. The question in the WHOQOL-BREF on sleep satisfaction seems to be suitable for this as a simple minimum variant for a sleep assessment.

### 4.2. Distribution of Sleep Satisfaction in Tinnitus

There was a relatively symmetrical distribution of (very) sleep-satisfied and (very) sleep-dissatisfied patients, with small proportions of patients who were neither satisfied nor dissatisfied with their sleep.

A relatively high proportion of patients with tinnitus (41.4%) who were satisfied or very satisfied with their sleep was observed. Notably this does not mean that these patients are free from sleep problems, e.g., increased falling asleep latency due to tinnitus. Instead, the indicated sleep satisfaction implies that the patients either have no sleep problems or can cope sufficiently with them. This thesis is supported by the fact that within the group of sleep-satisfied patients only the minority (19.7%) stated to be very satisfied with sleep. Possibly, the numerically largest group of “only” sleep-satisfied patients (see Figure 1: proportion of the sleep-satisfied group: 80.3%, proportion of the total sample: 33.2%) also includes those patients who have sleep problems due to tinnitus without these becoming subjectively relevant.

Concurrently, 40.9% of tinnitus patients reported being dissatisfied or very dissatisfied with their sleep. Within the group of sleep-dissatisfied patients, a more symmetrical subgroup distribution was found (see Figure 1: 65.1% dissatisfied resp. 34.9% very dissatisfied; proportion of the total sample: 26.6% resp. 14.3%). Thus, despite the relatively symmetrical distribution of the pooled groups, a shift towards sleep dissatisfaction is evident. If the prevalence of sleep satisfaction is used as a comparative measure for sleep complaints in patients with tinnitus, the prevalence of 40.1% is in the intermediate range of the large spectrum from 10.1% to 79.5% in different studies [24]. Since the present sample consists of help-seeking patients with a tendency towards higher disease burden, this may also mean that the mere prevalence of sleep problems in tinnitus patients is not a suitable measure of their functional relevance. This should be considered when selecting patients for targeted sleep interventions, so that the focus here should be on those patients in whom sleep disturbances have a functionally relevant effect.

### 4.3. Higher Burden in Sleep-Dissatisfied Patients: Depression, Tinnitus and Reduced Quality of Life

The group of sleep-dissatisfied patients showed a significantly higher burden in all questionnaires, but also regarding the presence of comorbidities.

Taking into account that dissatisfaction with sleep is related to an overall higher disease burden, the observed distribution provides empirical support for the recently proposed diagnostic differentiation between “tinnitus” (for the auditory percept) and “tinnitus disorder” (for the auditory percept plus associated impairments) [3].

Sleep-dissatisfied patients showed significantly higher scores in the MDI. Depressive symptoms in patients with tinnitus occur frequently and are estimated to be a major factor influencing distress [17,70], with a recent systematic review finding a median prevalence of depression in patients with tinnitus of 33% [71]. A longitudinal study demonstrated that an improvement in depressive symptoms was associated with an improvement in tinnitus severity [72]. Studies also show that the extent of depressive symptoms contributes significantly to tinnitus severity [73] and that tinnitus severity, increases the likelihood of the presence of depressive symptoms [74]. However, it is also debated whether the association of depression and tinnitus severity could be an artifact due to overlapping content in the respective questionnaires [75]. In particular, an EEG study in patients with tinnitus could show that distress and depressive symptoms were correlated with the activation of specific networks despite a certain overlap area [76]. A bidirectional relationship has been established for depression and sleep [38], so that the postulated poor sleep quality in the group of sleep-dissatisfied patients could be a compounding factor and, at the same time, depressive symptoms could be favored by sleep problems. Beyond complex clinical interactions of tinnitus, sleep problems, and depressive symptoms, a genome-wide association study revealed evidence that genetic factors may also play a role in the associations [77].

In the THI and TQ, the sleep-dissatisfied patients also showed significantly higher scores. Both questionnaires measure tinnitus distress, with the THI focusing on impact on daily life and the TQ focusing on aspects such as lifestyle, auditory perception, or perceived general health. More than a decade ago, it was postulated that coactivation of nonspecific stress networks plays an important role in the interaction of stress, tinnitus, and depression [78,79]. In turn, for the development of sleep complaints such as insomnia, it is discussed that stress-induced mechanisms such as disrupted cortical networks, dysregulation in the autonomic nervous system, and alterations in the hypothalamic-pituitary-adrenal (HPA) axis play a role. In this context, the so-called sleep reactivity in the sense of a vulnerability factor is of essential importance. This is the extent to which an individual reacts to stress exposure with disturbed sleep [80]. Hyperarousal plays a key role in the development and maintenance of insomniac complaints, as it accelerates activation of the sympathetic nervous system, and, subsequently, activity patterns of the limbic and autonomic areas of the brain can be observed [81]. These mechanisms can also be observed in tinnitus, which is why hyperarousal has also been discussed as the “common denominator” of tinnitus and insomnia [35]. However, it should be noted again that insomniac complaints cannot necessarily be derived from sleep dissatisfaction. Likewise, due to the study design, it remains unclear what is the cause or consequence or whether there is a reciprocal relationship between tinnitus distress and disturbed sleep.

In the WHOQOL-BREF to assess quality of life, all four domains (physical health, mental health, social connection, and environmental influences) showed poorer quality of life in the group of sleep-dissatisfied patients. This is consistent with previous studies showing that tinnitus distress is generally associated with a poorer quality of life [11,82,83,84]. Typically, sleep is also affected [20]. Furthermore, it could be shown that sleep disturbances in tinnitus have an important predictive function with regard to reduced quality of life [23]. Different specific sleep disorders are known to be associated with low general quality of life, which underlines the profound associate symptoms of disturbed sleep [85]. Moreover, the results of the present study are consistent with the differentiated concept of sleep satisfaction that individual, social, and environmental factors should be taken into account when assessing sleep satisfaction [64].

### 4.4. More Comorbidities in Sleep-Dissatisfied Patients

In the group of sleep-dissatisfied patients, comorbidities such as headaches, vertigo/dizziness, TMJ, or neck pain occurred significantly more frequently. These findings are consistent with results from a previous study showing that tinnitus patients with comorbid headaches were more often impaired by tinnitus and also more often suffered from other comorbidities such as vertigo, neck pain, or TMJ complaints [19]. The increased impairment of tinnitus patients with comorbid headaches can be interpreted as an additive effect on health-related quality of life. Similarly, TMJ disorder is known to be associated with increased tinnitus distress in tinnitus patients [86]. In addition, it was postulated that the more frequent occurrence of further—especially pain-related—comorbidities suggests an unspecific amplification of sensory signals and a diagnostic overlap with somatoform disorder [19]. Two decades ago, it was discussed that chronic tinnitus could be understood as an auditory phantom pain [87]. In the following years, there was a growing awareness that pain and tinnitus share pathophysiological similarities [88]. Meanwhile, overlapping networks of chronic pain and tinnitus have been identified at the neurobiological level [79,89], and it is assumed that a dysfunctional frontostriatal gating mechanism plays a role in the pathophysiology of both tinnitus and chronic pain [90]. Chronic pain is associated with impaired sleep [91], so the present data are consistent with previously reported data. Another mediating factor could be depressive symptomatology, considering that depression and sleep are bidirectionally related [38] and chronic pain and tinnitus are associated with depression [71,92]. In a recent therapy study, it was shown that a multimodal tinnitus-specific therapy program could improve both tinnitus distress and the affective pain component [93]. Overall, a complex interaction of pain-related comorbidities, impaired sleep, depressive symptoms, and tinnitus distress can be assumed.

### 4.5. Predictive Accuracy of the Model

Overall, the model obtained a modest accuracy of 69.5% (95% confidence interval of 65–73.6%), and a R^2^ (i.e., the coefficient of determination) of 26.0%. The weakness of the model was the poor predictive power of sleep satisfaction of patients who were neutral about sleep satisfaction. For the group of sleep-satisfied and sleep-dissatisfied patients, the accuracy was substantially better with 79.0% (with 87.8% sensitivity and 70.4% specificity) and 80.7% (with 83.0% sensitivity and 78.4% specificity), respectively. These data clearly demonstrate that sleep satisfaction is associated with several clinically relevant aspects of tinnitus and that high scores on the questionnaires are usually associated with sleep dissatisfaction. Sleep satisfaction can be assessed with a single question, allowing rapid identification of patients for whom differential sleep assessment is indicated. In addition, the single question on sleep satisfaction could also identify those patients at risk for co-occurring conditions that may also require further intervention.

### 4.6. Influencing Factors on the Model

From the coefficients of the regression model (see Table 2), it was observed that poor physical and psychological quality of life, depression, and tinnitus distress had the strongest effects on classifying patients according to their sleep satisfaction. Thus, these results are largely consistent with the distributional differences and underline the importance of these aspects in relation to tinnitus, which have been discussed in detail previously. Interestingly, epidemiological studies of the general population also show that self-reported stress and overall health were the strongest independent predictors of sleep satisfaction [63]. This suggests that the influencing factors are non-tinnitus specific and sleep disturbances in tinnitus could be conceived as a transdiagnostic process.

Furthermore, a possible gender effect was observed from the results of the Lasso regression: men were less likely to be dissatisfied with sleep than women. This is in line with previous studies, which could show that women suffered more from sleep disturbances [94,95]. A possible explanation for the sex difference could be the (peri-)menopause of women, in which sleep problems increase, and which could thus have contributed to the sex differences in the rating of sleep [96]. However, menopausal status was not assessed, so no definite conclusions can be drawn in this regard. Furthermore, the quality of tinnitus was found to influence sleep satisfaction. For example, non-pulsatile tinnitus and non-tonal tinnitus sounds were more frequently associated with sleep satisfaction (see Table 2). This is an unexpected finding and indicates for the first time a relationship between perceptual characteristics of tinnitus and sleep problems.

### 4.7. Limitations

In this study sleep satisfaction was evaluated in a large sample by one single question. No examination of subjective sleep quality was performed using validated self-report questionnaires. Similarly, no objective examinations such as actigraphy or polysomnography were used to quantify sleep problems Thus, we can only draw conclusions about subjective sleep satisfaction and not about objective sleep parameters.

Due to the cross-sectional design of the study, conclusions about causal relationships are not possible. Thus, it cannot be clarified whether the diverse investigated factors determine sleep satisfaction or whether sleep satisfaction determines investigated factors or whether sleep dissatisfaction and tinnitus are related via a third factor such as hyperarousal. In principle, a combination of the correlations would also be conceivable in the sense of a bidirectional relationship without this being able to be clarified on the basis of the present data.

It should also be noted that some common organic diseases such as chronic otitis media, Meniere’s disease, multiple otosclerosis, Alzheimer’s disease, and some other diseases in otorhinolaryngology or neurology could have an influence on subjective tinnitus as well as on auricular accompanying symptoms including hearing loss and otorrhea, which are also risk factors for sleep quality. However, these conditions were not systematically recorded in the present study.

Moreover, it should be considered that our sample comes from a specialized clinic and, therefore, is not representative for all people with tinnitus. Instead, the sample consists of help-seeking patients, and, therefore, will be biased towards a higher disease burden.

## 5. Conclusions

This cross-sectional study showed a multinomial distribution of sleep satisfaction versus sleep dissatisfaction in tinnitus patients. The division into sleep-satisfied and sleep-dissatisfied tinnitus patients allows a very good discrimination regarding disease burden. Sleep-dissatisfied patients were significantly more burdened in all questionnaires and suffered significantly more often from comorbidities. The significant group differences underscore the relevance of the concept of “tinnitus” versus “tinnitus disorder”, but also the importance of sleep satisfaction as a global health indicator. Given that common organic diseases influencing tinnitus and sleep have not been routinely considered so far, it is suggested that some more scientific and practical questionnaires will be updated and applied with the help of otolaryngologists or neurologists. Overall, the absence of sleep satisfaction could be identified as an important indicator for tinnitus distress and poor general health in tinnitus patients, although the data did not enable causal conclusions to be drawn. To address whether there is a causal or bidirectional relationship between tinnitus and sleep satisfaction, a random, double-blind, multicenter cohort study should be conducted in the future. Moreover, the assessment of sleep satisfaction with one single question represents a suitable indicator for the functional relevance of sleep disturbances in tinnitus patients, which is easily feasible in clinical routine.

## Figures and Tables

**Figure 1 ijerph-19-11005-f001:**
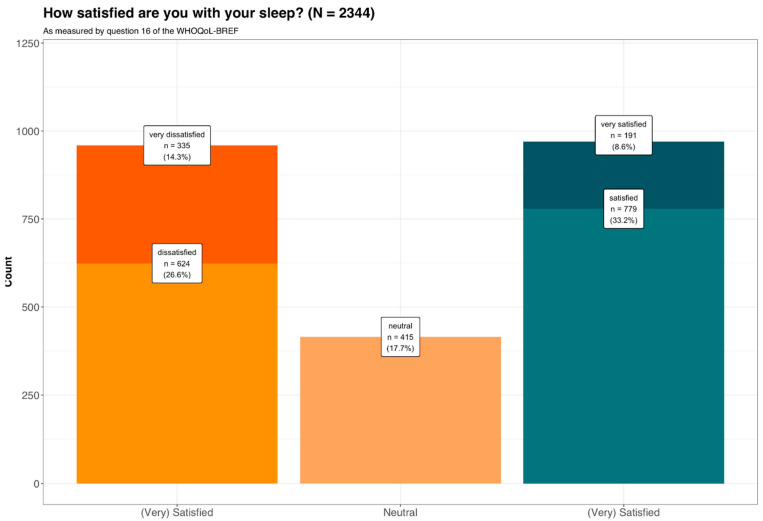
The distribution of sleep satisfaction. The groups very dissatisfied and dissatisfied with sleep, as well as the groups satisfied and very satisfied with sleep were grouped together yielding three groups that were used as response variable: (very) dissatisfied with sleep, neutral, and (very) satisfied with sleep.

**Table 1 ijerph-19-11005-t001:** Demographics of the study sample (*n* = 2344), stratified according to the patient’s reported sleep satisfaction as measured by question 16 of the Bref-WHOQOL (see methods). *p*-values were corrected for multiple comparisons using the Holm method [58].

Sample (N = 2344, 100.00%)	(Very) Dissatisfied (N = 959, 40.91%)	Neutral (N = 415, 17.70%)	(Very) Satisfied (N = 970, 41.38%)	Test Statistic (Degrees of Freedom)	*p*-Value
**Age**				H = 3.01 (2)	1
Mean (SD)	56.6 (12.4)	55.6 (13.2)	55.9 (13.7)		
Median [Min, Max]	57.0 [19.0, 87.0]	57.0 [0, 90.0]	56.0 [19.0, 92.0]		
**Sex**				H = 6.7 (2)	0.345
Female	366 (38.2%)	132 (31.8%)	327 (33.7%)		
Male	593 (61.8%)	283 (68.2%)	643 (66.3%)		
**MDI score**				H = 619.4 (2)	<0.001
Mean (SD)	21.0 (11.6)	14.1 (9.81)	8.94 (7.99)		
Median [Min, Max]	19.0 [1.00, 50.0]	11.0 [0, 48.0]	6.00 [0, 45.0]		
**THI score**				H = 599.9 (2)	<0.001
Mean (SD)	58.8 (22.0)	47.9 (21.4)	36.7 (20.5)		
Median [Min, Max]	60.0 [2.00, 100]	46.0 [4.00, 100]	34.0 [0, 96.0]		
Missing	6 (0.6%)	3 (0.7%)	10 (1.0%)		
**TQ Score**				H = 586.1 (2)	<0.001
Mean (SD)	49.2 (16.7)	39.9 (16.3)	31.6 (15.4)		
Median [Min, Max]	51.0 [2.00, 83.0]	40.0 [4.00, 80.0]	30.0 [0, 79.0]		
Missing	179 (18.7%)	73 (17.6%)	124 (12.8%)		
**Physical health (WHOQOL-BREF)**				H = 1262.1 (2)	<0.001
Mean (SD)	11.5 (1.60)	12.7 (1.45)	13.8 (1.26)		
Median [Min, Max]	11.0 [5.00, 17.0]	13.0 [9.00, 18.0]	14.0 [9.00, 18.0]		
Missing	7 (0.7%)	4 (1.0%)	3 (0.3%)		
**Psychological Health (WHOQOL-BREF)**				H = 401.7 (2)	<0.001
Mean (SD)	12.9 (2.16)	13.7 (1.89)	14.5 (1.68)		
Median [Min, Max]	13.0 [6.00, 18.0]	14.0 [7.00, 18.0]	15.0 [7.00, 18.0]		
Missing	7 (0.7%)	3 (0.7%)	4 (0.4%)		
**Social Factors (WHOQOL-BREF)**				H = 194.5 (2)	<0.001
Mean (SD)	13.7 (3.43)	14.6 (3.09)	15.5 (3.00)		
Median [Min, Max]	15.0 [4.00, 20.0]	15.0 [4.00, 20.0]	16.0 [4.00, 20.0]		
Missing	10 (1.0%)	0 (0%)	7 (0.7%)		
**Environment (WHOQOL-BREF)**				H = 291.5 (2)	<0.001
Mean (SD)	15.8 (2.31)	16.5 (2.08)	17.3 (1.92)		
Median [Min, Max]	16.0 [6.00, 20.0]	17.0 [10.0, 20.0]	18.0 [9.00, 20.0]		
Missing	4 (0.4%)	1 (0.2%)	0 (0%)		
**Family Member with Tinnitus**				χ^2^ = 0.3 (2)	1
yes	242 (25.2%)	102 (24.6%)	234 (24.1%)		
no	693 (72.3%)	303 (73.0%)	709 (73.1%)		
Missing	24 (2.5%)	10 (2.4%)	27 (2.8%)		
**Tinnitus initial Onset**				χ^2^ = 1.01 (2)	1
gradual	440 (45.9%)	183 (44.1%)	458 (47.2%)		
abrupt	467 (48.7%)	212 (51.1%)	470 (48.5%)		
Missing	52 (5.4%)	20 (4.8%)	42 (4.3%)		
**Pulsating Tinnitus**				χ^2^ = 22.9 (4)	0.002
yes, with the heartbeat	116 (12.1%)	47 (11.3%)	79 (8.1%)		
Yes, not with the heartbeat	113 (11.8%)	45 (10.8%)	72 (7.4%)		
no	704 (73.4%)	316 (76.1%)	801 (82.6%)		
Missing	26 (2.7%)	7 (1.7%)	18 (1.9%)		
**Tinnitus Location**				χ^2^ = 4.4 (6)	1
Unilateral	256 (26.7%)	119 (28.7%)	291 (30.0%)		
Bilateral	579 (60.4%)	247 (59.5%)	551 (56.8%)		
Head	103 (10.7%)	47 (11.3%)	107 (11.0%)		
Elsewhere	1 (0.1%)	0 (0%)	0 (0%)		
Missing	20 (2.1%)	2 (0.5%)	21 (2.2%)		
**Tinnitus Presentation**				χ^2^ = 12.2 (2)	0.033
intermittent	114 (11.9%)	77 (18.6%)	156 (16.1%)		
constant	832 (86.8%)	335 (80.7%)	798 (82.3%)		
Missing	13 (1.4%)	3 (0.7%)	16 (1.6%)		
**Loudness Fluctuation**				χ^2^ = 8.8 (2)	0.145
yes	612 (63.8%)	279 (67.2%)	571 (58.9%)		
no	330 (34.4%)	132 (31.8%)	378 (39.0%)		
Missing	17 (1.8%)	4 (1.0%)	21 (2.2%)		
**Subjective Loudness**				H = 144.3 (2)	<0.001
Mean (SD)	73.7 (64.5)	68.3 (68.9)	66.3 (84.4)		
Median [Min, Max]	70.0 [1.00, 999]	70.0 [5.00, 999]	60.0 [0, 999]		
Missing	39 (4.1%)	12 (2.9%)	43 (4.4%)		
**Type of Tinnitus**				χ^2^ = 6.1 (6)	1
tone	596 (62.1%)	277 (66.7%)	602 (62.1%)		
noise	120 (12.5%)	39 (9.4%)	107 (11.0%)		
crickets	135 (14.1%)	50 (12.0%)	149 (15.4%)		
others	83 (8.7%)	35 (8.4%)	85 (8.8%)		
Missing	25 (2.6%)	14 (3.4%)	27 (2.8%)		
**Tinnitus Pitch**				χ^2^ = 14.4 (6)	0.281
Very High Frequency	275 (28.7%)	108 (26.0%)	248 (25.6%)		
High Frequency	481 (50.2%)	219 (52.8%)	472 (48.7%)		
Medium Frequency	136 (14.2%)	59 (14.2%)	179 (18.5%)		
Low Frequency	18 (1.9%)	8 (1.9%)	33 (3.4%)		
Missing	49 (5.1%)	21 (5.1%)	38 (3.9%)		
**Sounds suppress tinnitus**				χ^2^ = 14.6 (4)	0.073
yes	594 (61.9%)	275 (66.3%)	675 (69.6%)		
no	201 (21.0%)	67 (16.1%)	156 (16.1%)		
not known	140 (14.6%)	69 (16.6%)	125 (12.9%)		
Missing	24 (2.5%)	4 (1.0%)	14 (1.4%)		
**Sounds worsens Tinnitus**				χ^2^ = 14.9 (4)	0.069
yes	554 (57.8%)	232 (55.9%)	519 (53.5%)		
no	210 (21.9%)	93 (22.4%)	277 (28.6%)		
not known	181 (18.9%)	88 (21.2%)	161 (16.6%)		
Missing	14 (1.5%)	2 (0.5%)	13 (1.3%)		
**Somatic Tinnitus**				χ^2^ = 5.3 (4)	1
yes	380 (39.6%)	155 (37.3%)	355 (36.6%)		
no	561 (58.5%)	256 (61.7%)	598 (61.6%)		
not known	3 (0.3%)	1 (0.2%)	8 (0.8%)		
Missing	15 (1.6%)	3 (0.7%)	9 (0.9%)		
**Effect of Naps on Tinnitus**				χ^2^ = 10.9 (6)	0.799
no effect	626 (65.3%)	277 (66.7%)	676 (69.7%)		
no info	38 (4.0%)	15 (3.6%)	30 (3.1%)		
reduces	81 (8.4%)	39 (9.4%)	101 (10.4%)		
worsens	174 (18.1%)	68 (16.4%)	131 (13.5%)		
Missing	40 (4.2%)	16 (3.9%)	32 (3.3%)		
**Sleep influences Tinnitus**				χ^2^ = 90.1 (4)	<0.001
yes	269 (28.1%)	88 (21.2%)	141 (14.5%)		
no	245 (25.5%)	128 (30.8%)	419 (43.2%)		
Not Known	408 (42.5%)	187 (45.1%)	379 (39.1%)		
Missing	37 (3.9%)	12 (2.9%)	31 (3.2%)		
**Effect of Stress on Tinnitus**				χ^2^ = 24.5 (4)	0.001
worsens	719 (75.0%)	296 (71.3%)	650 (67.0%)		
improves	10 (1.0%)	8 (1.9%)	7 (0.7%)		
no effect	195 (20.3%)	98 (23.6%)	285 (29.4%)		
Missing	35 (3.6%)	13 (3.1%)	28 (2.9%)		
**Hearing Difficulties**				χ^2^ = 0.03 (2)	1
yes	572 (59.6%)	248 (59.8%)	579 (59.7%)		
no	366 (38.2%)	157 (37.8%)	375 (38.7%)		
Missing	21 (2.2%)	10 (2.4%)	16 (1.6%)		
**Wears Hearing Aids**				χ^2^ = 6 (6)	<0.001
right	28 (2.9%)	11 (2.7%)	15 (1.5%)		
left	24 (2.5%)	8 (1.9%)	30 (3.1%)		
both	121 (12.6%)	55 (13.3%)	121 (12.5%)		
none	761 (79.4%)	332 (80.0%)	779 (80.3%)		
Missing	25 (2.6%)	9 (2.2%)	25 (2.6%)		
**Hyperacusis**				χ^2^ = 53.8 (8)	<0.001
never	76 (7.9%)	41 (9.9%)	115 (11.9%)		
rarely	106 (11.1%)	56 (13.5%)	151 (15.6%)		
sometimes	343 (35.8%)	158 (38.1%)	381 (39.3%)		
usually	172 (17.9%)	86 (20.7%)	174 (17.9%)		
always	248 (25.9%)	71 (17.1%)	137 (14.1%)		
Missing	14 (1.5%)	3 (0.7%)	12 (1.2%)		
**Pain from Noises**				χ^2^ = 53.5 (8)	<0.001
never	76 (7.9%)	41 (9.9%)	115 (11.9%)		
rarely	106 (11.1%)	56 (13.5%)	151 (15.6%)		
sometimes	343 (35.8%)	158 (38.1%)	381 (39.3%)		
usually	172 (17.9%)	86 (20.7%)	174 (17.9%)		
always	248 (25.9%)	71 (17.1%)	137 (14.1%)		
Missing	14 (1.5%)	3 (0.7%)	12 (1.2%)		
**Headaches**				χ^2^ = 69.5 (2)	<0.001
yes	443 (46.2%)	157 (37.8%)	275 (28.4%)		
no	492 (51.3%)	251 (60.5%)	681 (70.2%)		
Missing	24 (2.5%)	7 (1.7%)	14 (1.4%)		
**Vertigo/Dizziness**				χ^2^ = 35.7 (2)	<0.001
yes	371 (38.7%)	145 (34.9%)	256 (26.4%)		
no	555 (57.9%)	260 (62.7%)	688 (70.9%)		
Missing	33 (3.4%)	10 (2.4%)	26 (2.7%)		
**TMJ**				χ^2^ = 25.8 (2)	<0.001
yes	317 (33.1%)	110 (26.5%)	224 (23.1%)		
no	621 (64.8%)	296 (71.3%)	735 (75.8%)		
Missing	21 (2.2%)	9 (2.2%)	11 (1.1%)		
**Neck Pain**				χ^2^ = 58.4 (2)	<0.001
yes	623 (65.0%)	246 (59.3%)	469 (48.4%)		
no	314 (32.7%)	159 (38.3%)	482 (49.7%)		
Missing	22 (2.3%)	10 (2.4%)	19 (2.0%)		
**Psychiatric Treatment**				χ^2^ = 49.2 (2)	<0.001
yes	241 (25.1%)	88 (21.2%)	124 (12.8%)		
no	703 (73.3%)	322 (77.6%)	836 (86.2%)		
Missing	15 (1.6%)	5 (1.2%)	10 (1.0%)		

SD: standard deviation; Min: minimum; Max: maximum; MDI: Major Depression Inventory; THI: Tinnitus Handicap Inventory; TQ: Tinnitus Questionnaire; WHOQOL: World Health Organization quality of life brief version; TMJ: temporomandibular joint disorder; H: Kruskal–Wallis test; χ^2^ chi-squared test.

**Table 2 ijerph-19-11005-t002:** Coefficients of multinomial regression with Lasso regularization. Dashed lines represent predictors shrunk to 0 by the model. For categorical values, the log-odds of being classified into one of the three groups is compared to its reference, which is highlighted for each categorical variable.

	(Very) Dissatisfied	Neutral	(Very) Satisfied
**Age**	0.11		−0.03
**MDI Score**	0.35		−0.14
**THI Score**			−0.04
**TQ Score**	0.23		−0.15
**Physical Health (WHOQOL-BREF)**	−0.64		1.14
**Psychological Health (WHOQOL-BREF)**	0.18		−0.09
**Social Factors (WHOQOL-BREF)**			
**Environmental Factors (WHOQOL-BREF)**	0.05		
**Subjective Loudness (TSCHQ)**	0.00		
**Sex (reference: Female)**			
Male	−0.20		
**Family Member with Tinnitus (reference: Yes)**			
No			−0.04
**Tinnitus initial Onset (reference: Gradual)**			
Abrupt	−0.04	0.05	
**Pulsating Tinnitus (reference: yes, with the heartbeat)**			
Yes, not with the heartbeat			
No			0.37
**Tinnitus Location (reference: elsewhere)**			
Unilateral	−0.02		0.22
Bilateral			
Inside the head	−0.05	0.02	
**Tinnitus Presentation (reference: intermittent)**			
Constant	0.05	−0.10	
**Loudness Fluctuations (reference: yes)**			
No	0.05	−0.05	
**Type of Tinnitus (reference: tone)**			
Noise			0.14
Crickets		−0.10	0.14
Other			0.06
**Tinnitus Pitch (reference: low frequency)**			
High Frequency		0.01	
Medium Frequency	0.11		
**Sounds Suppress Tinnitus (reference: yes)**			
no		−0.03	
not known			
**Sounds Worsens Tinnitus (reference: yes)**			
no	0.12		
not known		0.13	−0.14
**Somatic Tinnitus (reference: yes)**			
no		0.03	−0.08
**Effect of Naps on Tinnitus (reference: not know)**			
no effect			
improves tinnitus	−0.10		0.02
worsens tinnitus	0.02		−0.21
**Sleep influences Tinnitus (reference: yes)**			
no	−0.10		0.34
not known	−0.23		
No effect	0.06		
**Hearing Difficulties (reference: yes)**			
no	0.01		−0.07
**Wears Hearing Aid (reference: one ear)**			
Both Ears		0.20	
None	0.01		−0.02
**Hyperacusis (reference: never)**			
Rarely	0.02		−0.03
Sometimes			0.03
Usually	−0.15		
Always			0.10
**Pain from Noises (reference: never)**			
Rarely			
Sometimes			
Usually	0.01		
Always		−0.01	
**Headaches (reference: yes)**			
No	−0.20		
**Vertigo (reference: yes)**			
No	0.11	−0.14	
**TMJ (reference: yes)**			
No	−0.05		
**Neck Pain (reference: Yes)**			
No	−0.11		0.18
**Psychiatric Treatment (reference: Yes)**			
No		−0.16	

MDI: Major Depression Inventory; THI: Tinnitus Handicap Inventory; TQ: Tinnitus Questionnaire; WHOQOL: World Health Organization quality of life brief version; TSCHQ: Tinnitus Sample Case History Questionnaire; TMJ: temporomandibular joint disorder.

**Table 3 ijerph-19-11005-t003:** Confusion matrix showing the results of the model accuracy in the test sample (*n* = 455).

	True Group		
Model Prediction	(Very) Dissatisfied	Neutral	(Very) Satisfied
(very) dissatisfied	151	37	22
Neutral	0	0	1
(very) satisfied	31	48	165

## Data Availability

The data presented in this study are available on reasonable request from the corresponding author.

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
