# Peer review of "Low Sleep Satisfaction Is Related to High Disease Burden in Tinnitus"

_ijerph, 2022, doi:10.3390/ijerph191711005_

Round 1

Reviewer 1 Report

This article aimed to evaluate the association between subjective sleep satisfaction and tinnitus-related disorders, which demonstrated that low sleep satisfaction was related to high disease burden in tinnitus. As authors said, sleep satisfaction with one single question was likely to represent a suitable and feasible indicator for evaluating sleep disturbances in clinical routine. However, in fact, some common organic diseases, such as chronic otitis media, Meniere’s disease, multiple otosclerosis, Alzheimer’s Disease and some others in Otorhinolaryngology or Neurology, could influence subjective tinnitus as well as auricular accompanying symptoms including hearing loss, otorrhea, which was also a risky factor for sleep quality. If possible, it’s recommended that some more scientific and practical questionnaires will be updated and applied with the help of Otorhinolaryngologist or Neurologist. Hopefully a random, double-blind, multiple-center cohort study will be conducted in future.

Author Response

Dear Reviewer,

Thank you very much for the valuable hints and comments. Please find our answers in the document attached.

Kind regards

The authors

Reviewer 2 Report

Note that the rating of "Average" for originality is related to the vast literature in the hearing sciences linking sleep and tinnitus. This reviewer is uncertain re: the degree to which such articles are routinely offered to the readership of IJERPH. Additional comments below:

The authors offer an analysis of sleep satisfaction among patients participating in a tinnitus clinic. The placement of sleep quality as a central element distinguishing “tinnitus” from “tinnitus disorder” is reasonable and consistent w/ current use in the literature. Although the effect of tinnitus on sleep is well-documented through decades of study in the hearing literature, the use of sleep as a variable to address tinnitus burden, as well as to identify patients at risk for substantial co-occurring conditions may support identification and management of patients among the readership of a journal not focused specifically on audiologic practice.

Perhaps the authors could comment on the potential effects of menopause on the female participants in the study. Given that the sex differences emerge from the analysis, it would be reasonable for the authors to comment on the possibility that menopause, or perimenopause, contributed to the sleep rating differences related to sex. Were ratings of sleep satisfaction influenced more by age among females such that pre-menopausal females experienced greater sleep satisfaction than the older participants? If nothing else, the issue merits mention in the discussion, perhaps in the vicinity of lines 412-413.

Introduction:

Line 50: suggest omitting “in the sense of a phantom sound” as the phrase does not add meaning to the sentence.

Line 65: consider change to, “…by patients with tinnitus and are a major factor reducing patients’ quality of life.” (or perhaps, “reducing patients’ quality of life ratings.”)

Lines 68-69: we have also observed tinnitus interfering with sleep in patients with trauma histories who experience nightmares and express that they are at times afraid to go to sleep.

Lines 82-83: suggest “reduction” rather than “improvement” as the latter could be interpreted as a worsening of the symptoms.

Line 84: omit “Conversely” as the sentence re: CBT agrees w/ the sentence immediately preceding.

Methods:

Line 178: were any criteria employed re: the 80/20 “training” and “test” samples? Assuming here that the split was random, but a brief mention would be welcome.

Line 209: this reviewer is not experienced with the use of the 10-fold cross-validated Lasso regression, however, is it not the case that a 10-fold approach specifies 90% of participants would be used for “training” and 10% “testing”? Is this an issue, given the 80/20 split mentioned earlier?

Results:

Line 200, 207-208 and elsewhere: for consistency, perhaps the authors should change “worse sleep satisfaction” to “sleep dissatisfaction”

Lines 203-205 suggest that tinnitus sound type was similar across the three groups, however on line 223, the type of sound is stated as influencing sleep satisfaction. This seeming inconsistency requires greater explanation.

Discussion:

Line 298: suggest “Concurrently,…” or “At the same time…” as opposed to “In tandem”

Line 377: omit “Already”

Line 401: would it be reasonable to add something to indicate that the single question regarding sleep could also designate those patients at risk for co-occurring conditions that might require intervention as well?

Line 440: Shouldn’t this read “Sleep dissatisfied patients…”?

Author Response

Dear Reviewer,

Thank you very much for the valuable hints and comments. Please find our answers in the document attached.

Kind regards,

The authors
